# Expression of Matrix Metalloproteinase-2,-7,-9 in Serum during Pregnancy in Patients with Pre-Eclampsia: A Prospective Study

**DOI:** 10.3390/ijerph192114500

**Published:** 2022-11-04

**Authors:** Ayibaota Bahabayi, Nan Yang, Tong Xu, Yuting Xue, Lijuan Ma, Xunke Gu, Yongqing Wang, Keke Jia

**Affiliations:** 1Department of Clinical Laboratory, Peking University Third Hospital, Beijing 100191, China; 2Department of Clinical Laboratory, Peking University People’s Hospital, Beijing 100033, China; 3Department of Blood Transfusion, Peking University Third Hospital, Beijing 100191, China; 4Department of Obstetrics and Gynecology, Peking University Third Hospital, Beijing 100191, China

**Keywords:** pre-eclampsia, matrix metalloproteinase-2, matrix metalloproteinase-7, matrix metalloproteinase-9, normal pregnancy

## Abstract

Background: Matrix Metalloproteinases (MMPs) have been found to have important roles in vascular pathology and may be involved in the occurrence of pre-eclampsia. In this study, the serum levels of MMP-2, -7, -9 in normal pregnant women and pre-eclampsia patients were analyzed to assess their predictive value. Methods: A total of 1563 pregnant women from Peking University Third Hospital, from February 2021 to October 2021, were enrolled. Serum samples were collected from patients one to three times, during the different trimesters. Among the 102 singleton pre-eclampsia patients, we collected samples from 33 patients in the first trimester (6–13 GW), 33 in the second trimester (14–28 GW), 41 in the third trimester (29–41 GW) and 28 after onset of pre-eclampsia. Samples from each trimester were collected before the onset of pre-eclampsia. Then we selected 35, 37, 43 and 25 samples from 124 healthy pregnant women by matching their age, BMI and gestational weeks, using these as the control groups. Serum levels of MMP-2, -7, -9 were detected by ELISA. The receiver operating characteristic (ROC) curve was used to evaluate their predictive value. Results: Except for the first trimester, MMP-2 and MMP-7 were significantly higher in the pre-eclampsia group (*p* < 0.5). Additionally, in the pre-eclampsia group, MMP-9 increased significantly in the first trimester and after the onset of pre-eclampsia but decreased significantly in the second and third trimesters (*p* < 0.5). The ROC curve indicated that MMP-9, MMP-2 and MMP-7 were the best indicators for predicting pre-eclampsia in the first, second and third trimesters, respectively. Conclusion: Increased MMP-2 and MMP-7 levels and a decreased MMP-9 level seem to be related to the pathogenesis of pre-eclampsia and are expected to be potential predictors of pre-eclampsia.

## 1. Introduction

Pre-eclampsia, a heterogeneous, multisystem disorder, is a serious threat to maternal and fetal safety, with an incidence of 2–8% globally and 2–16% in developing countries and it is also the third leading cause of maternal mortality [1,2].

The main pathogenesis of pre-eclampsia is an incomplete trophoblast invasion and a uterine spiral artery remodeling disorder, which leads to persistent placental ischemia and hypoxia, triggering a series of oxidative stress responses and an imbalance of pro-angiogenic and anti-angiogenic factors [3,4,5]. Studies have shown that matrix metalloproteinases (MMPs) are a group of important regulators of angiogenesis and uterine remodeling [6,7,8,9]. As gestation progresses, MMPs are involved in blastocyst implantation, spiral artery remodeling and placenta formation, and are associated with the development of pre-eclampsia [3,8,9,10]. 

MMPs are a group of endogenous proteolytic enzymes that can degrade different components of the extracellular matrix [6,7]. The first MMP (MMP-1) was identified in 1962 by Gross [11]. Now there are 28 kinds of MMPs known, and 25 of them are localized in the human body. MMPs share common structural domains, a hydrophobic signal peptide sequence, a propeptide region, a catalytic domain, a hinge region and a C-terminal hemopexin-like domain [12]. The catalytic region has two zinc ion (Zn^2+^) binding sites and one calcium ion (Ca^2+^) binding site. The activity of MMPs requires a Zn^2+^ and Ca^2+^ activation.

According to the different degradation substrates, MMPs can be divided into six categories, namely collagenase, gelatinase, stromelysins, membrane-type and matrilysins, amongst others [6,7]. MMPs are produced by a variety of tissues and cells, mainly secreted by proinflammatory cells and uterine placental cells, including fibroblasts, endothelial cells, macrophages, vascular smooth muscle, lymphocytes, trophoblasts and neutrophils [6,7]. 

There is increasing evidence that the dysregulation of MMPs has been associated with pre-eclampsia [8,13]. We were particularly interested in three members of the MMP family, namely MMP-2, MMP-7 and MMP-9. MMP-2 and MMP-9 belong to gelatinase, also named gelatinase A and gelatinase B, respectively, they mainly degrade collagen types I, II, III, VII, X and gelatin [6,7]. MMP-7, also named Matrilysin-1, mainly degrades several types of collagens (III, IV, V, IX, X, XI), proteoglycans, fibronectin, elastin and casein [6,7,14]. MMP-2 may play a major role during implantation, MMP-9 during the trophoblast invasion and MMP-7 is considered to be a predictor of late onset pre-eclampsia [3,13,14,15]. Changes in the levels and activities of MMPs can lead to a decreased invasion ability of trophoblast cells, collagen deposition, insufficient spiral artery remodeling, decreased uterine perfusion pressure, placental ischemia and hypoxia, a release of various bioactive factors and finally, to pre-eclampsia.

However, the existing studies only measured the levels of MMPs in part of the pregnancy and did not analyze MMP levels throughout the whole pregnancy, which in turn lacks a longitudinal consistency and has limitations in clinical applications [16,17,18]. To address these shortcomings, we detected the serum levels of MMP-2, MMP-7 and MMP-9 in pre-eclampsia patients as well as normal pregnant women throughout pregnancy by the enzyme-linked immunosorbent assay (ELISA). Receiver operating characteristic (ROC) curve analysis was used to investigate the predictive values of these factors for pre-eclampsia.

## 2. Materials and Methods

### 2.1. Study Population

Approved by the Medical Ethics Committee of Peking University Third Hospital (No. IRB00006761-M2021032), a total of 1563 pregnant women, aged 18 to 45 years who underwent regular prenatal care at Peking University Third Hospital from February 2021 to October 2021, were enrolled. The remaining serum from routine biochemical tests from regular prenatal check-ups were collected and stored below −70 °C. Serum samples were collected from patients one to three times during the different trimesters. Meanwhile, we recorded the participants’ general clinical data and followed their pregnancy outcomes. Among the 1563 pregnant women, 333 patients who did not meet the study criteria were excluded, 102 women were eventually diagnosed with pre-eclampsia and 776 healthy pregnant women, without pre-eclampsia and other complications, were identified.

Among the 102 singleton pre-eclampsia patients, samples from 33 patients in the first trimester (6–13 GW), 33 in the second trimester (14–28 GW), 41 in the third trimester (29–41 GW) and 28 after onset of pre-eclampsia, were eventually collected. Samples from each trimester were collected before the onset of pre-eclampsia. Then we selected 35, 37, 43 and 25 samples from 124 healthy pregnant women by matching age, BMI and gestational weeks as a control group (Figure 1).

A total of 1563 pregnant women were enrolled and 333 pregnant women who did not meet the study criteria were excluded. Serum samples were collected from patients, one to three times during the different trimesters. Among the 102 singleton pre-eclampsia patients, we collected samples from 33 patients in the first trimester (6–13 GW), 33 in the second trimester (14–28 GW), 41 in the third trimester (29–41 GW) and 28 after onset of pre-eclampsia. Samples at each trimester were collected before the onset of pre-eclampsia. Then we selected 35, 37, 43 and 25 samples from 124 healthy singleton pregnant women by matching age, BMI, and gestational weeks as the control group.

### 2.2. Inclusion and Exclusion Criteria

The inclusion criteria for the study group were:Pre-eclampsia group: A singleton pregnancy that developed pre-eclampsia. The diagnosis of pre-eclampsia was based on the diagnostic criteria of ACOG practice bulletin No. 222 [19];Control group: singleton pregnancy, normal conception, normal blood pressure, no diabetes, nephropathy and/or other medical and surgical diseases and no autoimmune diseases.

Criteria for the exclusion from the study were: pregnancy after assisted reproductive technologies, multiple pregnancy, fetuses with abnormalities, fetuses with chromosomal abnormalities, stillbirths, neoplastic diseases, autoimmune diseases, infectious diseases (HIV infection, hepatitis), administration of medications and smoking or drug addictions.

### 2.3. Sample Collection

During regular prenatal care, 4 mL of fasting cubital venous blood was drawn from the pregnant women for routine biochemical testing. We collected samples without hemolysis, jaundice, or lipemia after testing. They were centrifuged at 2793 g for 10 min, and at least 500 μL of the upper serum was collected. Serum samples were stored in the refrigerator at −80 °C until testing and repeated freezing and thawing were avoided.

### 2.4. ELISA

Serum concentrations of MMP-2, MMP-7 and MMP-9 were determined by using an ELISA method according to the manufacturer’s instruction. Kits were purchased from R & D Systems (Catalogue NO. MMP200, DMP700, DMP900; Lot NO. P282479, P277063, P233697). 

### 2.5. Statistical Analysis

The measurement data were expressed as the median (25%, 75%) for non-normally distributed data. The Mann–Whitney U test was used to analyze the maternal characteristics data and MMP concentrations between the pre-eclampsia and control groups. A non-parametric test (Kruskal–Wallis) was used to analyze serum MMP-2, -7, -9 concentrations at different trimesters of each group. The concentrations of MMP-2, MMP-7 and MMP-9 in each subgroup (mild pre-eclampsia vs. severe pre-eclampsia, term pre-eclampsia vs. preterm pre-eclampsia) were also analyzed using the Mann–Whitney U test. The optimal cut-off value, specificity and sensitivity of MMP-2, -7, -9, for predicting pre-eclampsia were determined by the receiver operating characteristic curve (ROC). Each statistical test was two-tailed, and *p* values of less than 0.05 were considered statistically significant.

Statistical analysis was performed with SPSS software (version 26; SPSS Inc., Chicago, IL, USA). Graph Pad Prism 9.0 software (GraphPad, La Jolla, CA, USA) was used for plotting. 

## 3. Results

1.Clinical Characteristics analysis of the study groups.

Maternal characteristics of the study population are shown in Table 1. There were no statistical differences between the pre-eclampsia and control groups in terms of age, and body mass index (BMI) (*p* > 0.05). Systolic blood pressure (SBP), diastolic blood pressure (DBP) and mean arterial pressure (MAP) were significantly higher in the pre-eclampsia group than in the control group from the second trimester (*p* < 0.05). We also listed three high-risk histories of pre-eclampsia: eclampsia, gestational diabetes mellitus (GDM) and kidney disease. More than one third of women with pre-eclampsia had a history of GDM. In addition, information on the subgroups is also presented in Table 1. The number of patients with severe pre-eclampsia and preterm pre-eclampsia was less than the number of patients with mild pre-eclampsia and term pre-eclampsia.

2.Serum levels of MMP-2, MMP-7 and MMP-9 in pre-eclampsia and control groups during pregnancy

In the first trimester: The concentrations of MMP-2 and MMP-7 were lower in the pre-eclampsia group than in the control group, but these differences were statistically insignificant (*p* > 0.05). However, the concentration of MMP-9 in the pre-eclampsia group (991.88 (841.81, 1075.35))ng/mL was significantly higher than that in the control group (845.64 (747.24, 972.36)) ng/mL (*p* < 0.05) (Table 2 and Figure 2).

Serum concentrations of MMP-2, MMP-7 and MMP-9 in pre-eclampsia and normal pregnant women. The box plot shows the median (bold horizontal line), interquartile range (box) and total range (whiskers). *p* < 0.05 was considered statistically significant. ^ns^, *p* ≥ 0.05; * *p* ˂ 0.05; ** *p* ˂ 0.01; *** *p* ˂ 0.001.

In the second trimester: The level of MMP-2 in pre-eclampsia group [302.43 (177.69, 351.33)] ng/mL was significantly higher than that in control [228.64 (213.12, 254.7)] ng/mL (*p* < 0.05). The level of MMP-7 in the pre-eclampsia group [2.38 (1.87, 2.85)] ng/mL was also significantly higher than that in control [1.84 (1.38, 2.74)] ng/mL (*p* < 0.05). However, the level of MMP-9 was lower in pre-eclampsia group [968.15 (841.81, 1185)] ng/mL relative to control group [1220.06 (909.19, 1285.15)] ng/mL (*p* < 0.05) (Table 2 and Figure 2).

In the third trimester: The MMP-2 concentration in pre-eclampsia group [337.2 (279.12, 376.4)] ng/mL was significantly higher than that in control [282.52 (236.4, 344.01)] ng/mL (*p* < 0.05). The MMP-7 concentration in pre-eclampsia group [3.43 (2.39, 5.82)] ng/mL was also significantly higher than that in control [2.56 (1.65, 3.38)] ng/mL (*p* < 0.05). The MMP-9 concentration in pre-eclampsia group [1020.7 (880.71, 1170.25)] ng/mL was significantly lower than that in control [1263 (951.29, 2618.22)] ng/mL, (*p* < 0.05) (Table 2 and Figure 2).

After the onset of pre-eclampsia: The level of MMP-2 in pre-eclampsia group [302.85 (232, 342)] ng/mL was significantly higher than that in control [242 (217.8, 281.2)] ng/mL (*p* < 0.05). The level of MMP-7 in pre-eclampsia group [4.14 (3.37, 6.32)] ng/mL was significantly higher than that in control [3.48 (1.91, 4.16)] ng/mL, and the MMP-9 level in pre-eclampsia group [1066.92 (829.73, 1305.75)] ng/mL was also significantly higher than that in control [806 (581.47, 1051)] ng/mL, (*p* < 0.05) (Table 2 and Figure 2).

The results of the pairwise comparison within the groups are shown in Table 2. In the comparisons within the control group, it was found that there were significant differences in the three types of MMPs, while in the pre-eclampsia group, only MMP-7 showed partial differences.

3.Analysis of the ROC curve for MMP-2, MMP-7 and MMP-9

In the first trimester, the area under the curve (AUC) of MMP-9 was the largest among the three MMPs, and the cut-off value was ≥1003.752 ng/mL, which predicted the development of pre-eclampsia, with a sensitivity of 80% and a specificity of 88.6% (AUC = 0.722). In the second trimester, the prediction results of the three MMPs were similar, among which MMP-2 was the best. The performed ROC analysis allowed for determining the level of MMP-2 ≥ 294.987 ng/mL as the threshold concentration for predicting pre-eclampsia, with a sensitivity of 60.6% and a specificity of 97.3% (AUC = 0.663). In the third trimester, the AUC values of the three MMPs were all significant, among which MMP-7 was the best. When the cut-off value was ≥3.53 ng/mL, the sensitivity and specificity of the pre-eclampsia prediction were 50% and 80.6%, respectively (AUC = 0.675). After the onset of pre-eclampsia, the best diagnostic indicator was MMP-7. The ROC analysis showed that MMP-7 ≥ 4.07 ng/mL, was the threshold concentration for diagnosing pre-eclampsia, with a sensitivity of 58.6% and a specificity of 76.7% (AUC = 0.711) (Table 3 and Figure 3).

ROC, receiver operating characteristic; MMP-2, matrix metalloproteinase-2; MMP-7, matrix metalloproteinase-7; MMP-9, matrix metalloproteinase-9.

## 4. Discussion

Pre-eclampsia is a pregnancy complication with complex manifestations, variable symptoms, and rapid progression, which seriously threatens maternal and infant health [20]. Early identification of suspected pre-eclampsia patients is the key to reducing the impact of pre-eclampsia on the mother and fetus, as well as reducing the incidence of pre-eclampsia.

The pathogenesis of the disease has not been completely clarified. Studies have found that MMPs are involved in the vascular remodeling during a normal pregnancy, and the changes in their levels and activity may be one of the reasons for the development of pre-eclampsia [6,21,22].

In this study, we analyzed the changes of serum levels of MMP-2, -7 and -9 in the pre-eclampsia and control groups to evaluate their values to predict the risk of developing pre-eclampsia.

We found that the blood pressure of the pre-eclampsia groups were significantly higher than those of the control groups from the second trimester. This reflects a vascular remodeling disorder in patients with pre-eclampsia and suggests the importance of closely monitoring blood pressures [23]. In addition, more than one-third of the patients had a history of GDM, which can increase the chances of pre-eclampsia in pregnant women suggesting the necessity of nutritional management during pregnancy [24].

Our results were similar with some studies. Feng et al. compared plasma MMP-2 and MMP-9 levels between pre-eclampsia and control groups at 12–18 weeks and found that the level of MMP-2 was increased but was not statistically significant, while the level of MMP-9 was significantly decreased in the pre-eclampsia group [25]. Remarkably, the authors found that the ratio of MMP-2/MMP-9 was significantly higher in the pre-eclampsia group when compared with the control group, and this was considered as a more accurate biomarker for predicting pre-eclampsia. Chen Yimin et al. found that the serum level of MMP-9 in the pre-eclampsia group was significantly lower than that in the control group in the second and third trimesters, which was also consistent with the results of this study [26]. The study of He Hui et al. showed that compared with the control group, the serum level of MMP-2 was significantly increased and the level of MMP-9 was significantly decreased in the pre-eclampsia group after the disease onset [27]. Moreover, Elena Timokhina et al. found that the level of MMP-2 in the pre-eclampsia group was significantly higher than that in the control group [28]. However, when comparing the MMP-2 levels between the early-onset pre-eclampsia and late-onset pre-eclampsia, no significant differences were observed. In addition, compared with the control group, the level of MMP-9 in the pre-eclampsia group was decreased, and the level of MMP-9 in the early-onset pre-eclampsia group was lower than that in the late-onset pre-eclampsia group. Erez et al. found that the level of MMP-7 was elevated in pre-eclampsia patients at 8–22 gestational weeks and that this could be used as a predictor of late-onset pre-eclampsia [14].

These results suggest that the abnormal expression of MMP-2, -7, -9 is related to the pathogenesis of pre-eclampsia [10]. In the first trimester, high level of MMP-9 can destroy the integrity of vascular basement membranes and mediates endothelial injury thus allowing inflammatory cells to invade the injured tissue [29]. In the second and third trimesters, the increased MMP-2 and MMP-7 can accelerate the lysis of vasoactive peptide, which makes the patient’s vasodilatation function an obstacle thus increasing the vasodilatation function [14,30,31]. Therefore, we found that the blood pressure of pre-eclampsia patients was already significantly increased in the second trimester. The decrease in serum MMP-9 decreases trophoblast invasions, decreases collagen hydrolysis and increases collagen deposition, which in turn leads to a placental “shallow implantation” and insufficient spiral artery remodeling, all of which participate in the occurrence of pre-eclampsia [32].

Moreover, we found that the serum levels of MMP-2, MMP-7 and MMP-9 changed significantly between different trimesters within the control group, which suggest that the serum levels of MMPs are precisely regulated in normal pregnancies [32]. The peak expression of MMP-2 and MMP-7 in serum was in the first trimester, indicating that MMP-2 and MMP-7 were involved in the formation of the early placenta. The second trimester is a critical period for the formation of the fetus, at which time the fetus has a high demand for nutrition and oxygen [33]. If the MMP levels increase, they will promote various proteolysis, which will affect the maternal blood and oxygen supply and the formation of the fetus. Therefore, the low expression levels of MMPs were in the second trimester. The levels of MMPs increased in the third trimester, which may be related to the promotion of fetal maturation and placental detachment by MMPs. The expression of MMP-9 was lowest in the first trimester and highest in the second and third trimesters. The possible reason is that both MMP-9 and MMP-2, as gelatinases, have a mutually regulating and complementary relationship. MMP-2 plays a major role in the first trimester and MMP-9 plays a leading role in the second and third trimesters [15], but the mechanistic relationship between the two remains to be studied.

In the pre-eclampsia group, the expression levels of MMP-2 and MMP-9 were not statistically different throughout the three trimesters, while the expression level of MMP-7 was different at some periods. Throughout pregnancy, the expression of serum MMP-2 and MMP-7 in pre-eclampsia patients had been maintained at a high level, while the expression of MMP-9 was the contrary. These changes suggested that the regulation of serum MMP levels in pre-eclampsia patients was obviously out of balance, which is closely related to the pathogenesis of pre-eclampsia [10,29]. 

We analyzed changes in MMP-2, -7, -9 levels in the subgroups of the pre-eclampsia group. The number of patients with severe pre-eclampsia and preterm pre-eclampsia was less than the number of patients with mild pre-eclampsia and term pre-eclampsia. There were no significant statistical differences in the levels of the three MMPs between the mild group and the severe group. However, the MMP-7 level in the preterm group was significantly higher than that in the term group but only in the third trimester. Due to the small numbers of subgroup cases, the results may not be reliable.

The ROC results show that MMP-9, MMP-2 and MMP-7 had the best effect in predicting pre-eclampsia in the first, second and third trimesters, respectively. Moreover, MMP-7 showed the best diagnostic value for pre-eclampsia. 

The well-known predictive biomarkers for pre-eclampsia are sFlt-1 and PIGF, but their applications are still limited. Low PIGF after 22 gestational weeks is the strongest predictor for the subsequent development of late-onset pre-eclampsia [14]. Therefore, further biomarkers are needed to improve the accuracy of pre-eclampsia prediction in early pregnancy. With the rapid development of etiological research on pre-eclampsia, more etiological related indicators have been found, and predictive models are gaining more attention. We believe that the study of MMPs will help to improve the performance of the prediction model, and the mechanistic study of pre-eclampsia.

## 5. Strengths and Limitations

The main strength of this study is its prospective design. In addition, there are few reports into the changes of serum MMP levels in pre-eclampsia and control groups throughout pregnancy.

The limitations of this experiment are as follows: (1) The incidence of pre-eclampsia is low making it difficult to collect samples; therefore, the number of enrolled samples was limited; (2) The samples collected in the first trimester were concentrated in the 6–10 week range, which was insufficient to reflect the changes of serum MMP levels in the whole of the first trimester; (3) We did not analyze the efficacy of MMPs in combination with other indicators to predict pre-eclampsia.

In future studies, it is necessary to increase the sample size to obtain more data and increase the reliability of the statistical results. Meanwhile, the pre-eclampsia group can be subdivided into further subgroups for a more detailed study. In addition, some studies can be done to predict pre-eclampsia by combining MMPs with sFlt-1 and PIGF.

## 6. Conclusions

In conclusion, serum levels of MMP-2, -7, and -9 were significantly different between the pre-eclampsia and control groups during pregnancy. The increased MMP-2 and MMP-7 levels and a decreased MMP-9 level seemed to be related to the pathogenesis of pre-eclampsia and are expected to be potential predictors for developing pre-eclampsia.

## Figures and Tables

**Figure 1 ijerph-19-14500-f001:**
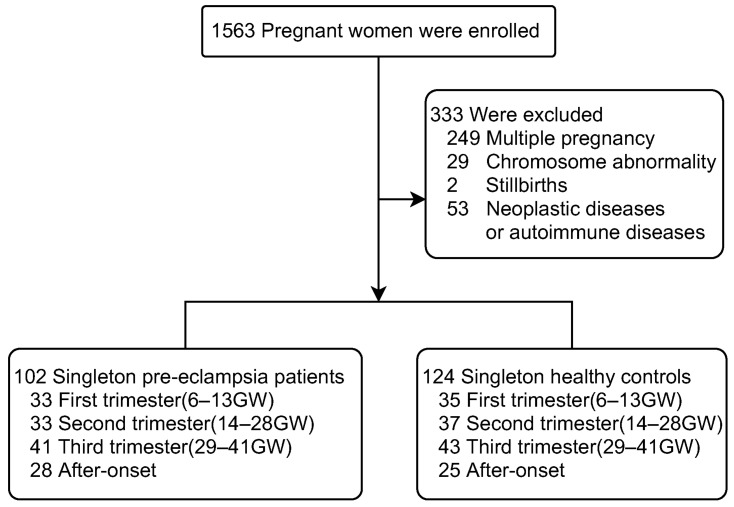
Numbers of pregnant women enrolled and outcomes of the study. GW, gestational age; BMI, body mass index.

**Figure 2 ijerph-19-14500-f002:**
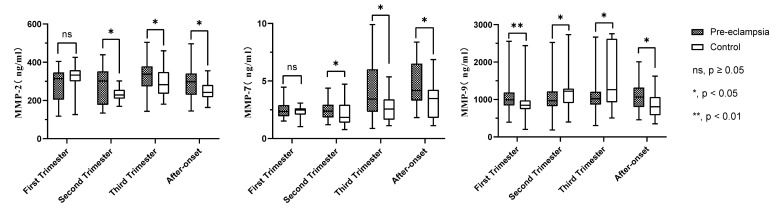
Box plots of MMP-2, MMP-7 and MMP-9 in pre-eclampsia and control groups during pregnancy. MMP-2, matrix metalloproteinase-2; MMP-7, matrix metalloproteinase-7; MMP-9, matrix metalloproteinase-9.

**Figure 3 ijerph-19-14500-f003:**
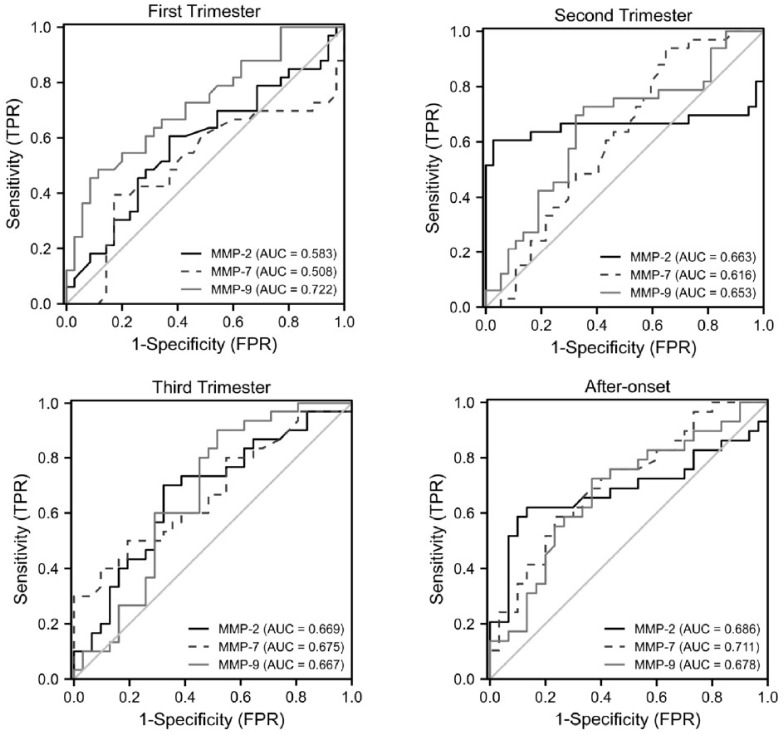
Analysis of the ROC curve for MMP-2, MMP-7 and MMP-9.

**Table 1 ijerph-19-14500-t001:** Clinical Characteristics of pre-eclampsia group and control group.

	Pre-Eclampsia Group (n = 135)				Control Group (n = 140)			
Parameters	First Trimester (n = 33)	Second Trimester (n = 33)	Third Trimester (n = 41)	After-Onset (n = 28)	First Trimester (n = 35)	Second Trimester (n = 37)	Third Trimester (n = 43)	After-Onset (n = 25)
Age (years)	34 (32, 38)	32 (31, 34)	33 (31, 36)	33 (31, 35)	33 (31, 36.5)	33 (31, 35)	33 (30, 35.5)	34 (31, 37)
BMI (Kg/m^2^)	23.83 (22.66, 27.79)	23.14 (21.77, 25.71)	23.44 (22.04, 24.92)	23.14 (20.83, 24.22)	23.31 (20.85, 26.67)	22.94 (20.03, 26.95)	23.11 (21.09, 23.84)	22.49 (20.96, 23.79)
Nulliparity (n (%))	27 (81.8%)	27 (81.8%)	34 (82.9%)	20 (71.4%)	24 (68.6%)	27 (73%)	26 (60.5%)	12 (48%)
Blood Pressure (mmHg)								
SBP	127 (119, 129)	126.5 (123, 134.25)	143 (140, 153.75)	140 (136.5, 154)	122 (114, 128)	121 (116, 126)	126 (123, 133.5	125 (124, 131)
DBP	74 (68, 82)	74.5 (68, 79.25)	90 (85.25, 96.75)	90 (85.5, 98.5)	70 (66, 76)	67 (63, 74)	72 (67, 79)	76 (71, 83)
MAP	92.67 (84.67, 97.67)	91.17 (87.5, 96.08)	106.67 (104.83, 115.17)	107.33 (101, 117.7)	87.33 (83, 94.67)	86 (80.33, 90)	90 (86.17, 97.17)	92.33 (87, 97.53)
*p* values of blood pressure								
P_s_	0.329	0.002	0.000	0.000				
P_d_	0.163	0.004	0.000	0.000				
P_m_	0.154	0.001	0.000	0.000				
History (n (%))								
eclampsia	0	5 (15.2%)	2 (4.9%)	3 (10.7%)	0	0	0	0
GDM	11 (33.3%)	13 (39.4%)	18 (43.9%)	10 (35.7%)	0	0	0	0
kidney disease	1 (3.0%)	2 (6.1%)	2 (4.9%)	1 (3.6%)	0	0	0	0
Number of cases in subgroups (n (%))								
Sever	11 (33.33%)	13 (39.40%)	18 (43.90%)	12 (42.86%)	0	0	0	0
Mild	22 (66.67%)	20 (60.60%)	23 (56.10%)	16 (57.14%)	0	0	0	0
Preterm	2 (6.06%)	4 (12.12%)	10 (24.39)	12 (42.86%)	7 (20%)	4 (10.81%)	5 (11.63%)	8 (32%)
Term	31 (93.94%)	29 (87.88%)	31 (75.61%)	16 (57.14%)	28 (80%)	33 (89.19%)	38 (88.37%)	17 (68%)

n, sample size; BMI, body mass index; SBP, systolic pressure; DBP, diastolic pressure; MAP, mean arterial pressure; MAP = (SBP + 2 × DBP)/3; GDM, gestational diabetes mellitus. Data are expressed as median (25%, 75%) or number (percentage). P_s_, the significance of the difference in SBP between the pre-eclampsia group and the control group; P_d_, the significance of the difference in DBP between the pre-eclampsia group and the control group; P_m_, the significance of the difference in MAP between the pre-eclampsia group and the control group. *p* < 0.05 was considered statistically significant.

**Table 2 ijerph-19-14500-t002:** Expression of MMP-2, MMP-7 and MMP-9 in pre-eclampsia and control groups during pregnancy.

Indicators	Gestation	Pre-Eclampsia Group (n = 111)	Control Group (n = 144)	Z	P_0_	P_1_	P_2_
MMP-2 (ng/mL)	First Trimester	314.44 (229.6, 346.35)	332.8 (304.41, 356.98)	−1.154	0.249	0.146	0.000 ^a^ 0.795 ^b^ 0.001 ^c^ 0.011 ^d^ 1.000 ^e^ 0.138 ^f^
	Second Trimester	302.43 (177.69, 351.33)	228.64 (213.12, 254.7)	2.053	0.040		
	Third Trimester	337.2 (279.12, 376.4)	282.52 (236.4, 344.01)	2.272	0.023		
	After-onset	302.85 (232, 342)	242 (217.8, 281.2)	2.585	0.010		
MMP-7 (ng/mL)	First Trimester	2.35 (1.94, 2.88)	2.48 (2.13, 2.62)	−0.045	0.964	1.000 ^a^ 0.009 ^b^ 0.000 ^c^ 0.014 ^d^ 0.000 ^e^ 0.553 ^f^	0.000 ^a^ 0.795 ^b^ 0.001 ^c^ 0.011 ^d^ 1.000 ^e^ 0.138 ^f^
	Second Trimester	2.38 (1.87, 2.85)	1.84 (1.38, 2.74)	2.001	0.045		
	Third Trimester	3.43 (2.39, 5.82)	2.56 (1.65, 3.38)	2.345	0.019		
	After-onset	4.14 (3.37, 6.32)	3.48 (1.91, 4.16)	2.350	0.019		
MMP-9 (ng/mL)	First Trimester	991.88 (841.81, 1075.35)	845.64 (747.24, 972.36)	3.148	0.002	0.979	0.000 ^a^ 0.000 ^b^ 1.000 ^c^ 1.000 ^d^ 0.002 ^e^ 0.000 ^f^
	Second Trimester	968.15 (841.81, 1185)	1220.06 (909.19, 1285.15)	−2.194	0.028		
	Third Trimester	1020.7 (880.71, 1170.25)	1263 (951.29, 2618.22)	−2.156	0.031		
	After-onset	1066.92 (829.73, 1305.75)	806 (581.47, 1051)	2.227	0.026		

n, sample size; MMP-2, matrix metalloproteinase-2; MMP-7, matrix metalloproteinase-7; MMP-9, matrix metalloproteinase-9; P_0_, pre-eclampsia group vs. control group; P_1_, *p* values for pairwise comparisons between different trimesters within the pre-eclampsia group; P_2_, *p* values for pairwise comparisons between different trimesters within the control group; ^a^, First Trimester vs. Second Trimester; ^b^, First Trimester vs. Third Trimester; ^c^, First Trimester vs. After-onset; ^d^, Second Trimester vs. Third Trimester; ^e^, Second Trimester vs. After-onset; ^f^, Third Trimester vs. After-onset. *p* < 0.05 was considered statistically significant.

**Table 3 ijerph-19-14500-t003:** Analysis of the ROC curve for MMP-2, MMP-7 and MMP-9.

Indicators	AUC	95% CI	Cut-Off (ng/mL)	Sensitivity (%)	Specificity (%)	PPV (%)	95% PPV	NPV (%)	95% NPV	Youden Index	*p*
First Trimester											
MMP-2	0.583	0.444–0.722	329.832	60.6	62.9	60.6	44–77	62.9	47–79	0.235	0.241
MMP-7	0.508	0.362–0.653	2.01	68.4	82.9	68.4	53–84	59.2	43–75	0.223	0.912
MMP-9	0.722	0.601–0.843	1003.752	80.0	88.6	80.0	66–94	64.6	49–80	0.371	0.002
Second Trimester											
MMP-2	0.663	0.507–0.819	294.987	60.6	97.3	95.2	78–99	73.5	59–88	0.579	0.019
MMP-7	0.616	0.483–0.749	1.53	93.9	35.1	56.4	39–73	86.7	76–98	0.291	0.096
MMP-9	0.653	0.521–0.785	1078.844	72.7	64.9	64.9	49–81	72.7	58–87	0.376	0.028
Third Trimester											
MMP-2	0.669	0.531–0.808	298.673	70.0	67.7	67.7	53–82	70.0	56–84	0.377	0.023
MMP-7	0.675	0.538–0.811	3.53	50.0	80.6	71.4	58–85	62.5	48–77	0.306	0.019
MMP-9	0.667	0.526–0.807	1315.878	90.0	48.4	62.8	48–78	83.3	72–94	0.384	0.025
After-onset											
MMP-2	0.686	0.538–0.833	289.2	62.1	86.7	81.8	68–96	70.3	52–88	0.487	0.014
MMP-7	0.711	0.579–0.843	4.07	58.6	76.7	70.8	54–88	65.7	47–84	0.353	0.005
MMP-9	0.678	0.538–0.817	881.5	72.4	63.3	65.6	48–83	70.4	53–88	0.357	0.019

MMP-2, matrix metalloproteinase-2; MMP-7, matrix metalloproteinase-7; MMP-9, matrix metalloproteinase-9; ROC, receiver operating characteristic; AUC, area under the ROC curve; CI, confidence interval; PPV, positive predictive value; NPV, negative predictive value.

## Data Availability

The data presented in this study are openly available in FigShare at [https://doi.org/10.6084/m9.figshare.21433605].

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
