# Peer review of "Expression of Matrix Metalloproteinase-2,-7,-9 in Serum during Pregnancy in Patients with Pre-Eclampsia: A Prospective Study"

_ijerph, 2022, doi:10.3390/ijerph192114500_

Round 1

Reviewer 1 Report

Abstract

Line 18: Do not underline “February” and decrease font size.

Lines 18-20: Statement “Including 33, 36, 42, 29 … to 35, 37, 43 and 29 healthy controls” not clear. Consider rephrasing.

Introduction

Line 39: Change “Matrix” to “matrix”.

Line 45: Insert a reference for Gross’s work

Line 60: Change “plays” to play.

Results

Table 3: Authors should indicate the p-values obtained after the ROC curve analysis.

Lines 129-131: Remove underlines from footnotes under Table 1.

Major Comments

Results

Lines 169-170: Authors indicated that after the onset of PE, the MMP-9 level in PE group [1066.92 169 (829.73,1305.75)] ng/ml was significantly lower than that in control [806.00 170 (581.47,1051.00)] ng/ml, (P<0.05). This contradicts the values obtained and findings from Fig 1.

Discussion

Lines 258-261: Authors should clarify the MMP-9 level in PE group and controls as suggested above, and consider rephrasing statements in the discussion.

Author Response

Dear Reviewer,

The title of the manuscript is Expression of matrix metalloproteinase-2、-7、-9 in serum during pregnancy in patients with pre-eclampsia: a prospective study. Thank you very much for your modification suggestions. The following is the modification I have made in response to your modification comments.

Abstract

Line 18: Do not underline “February” and decrease font size.

Lines 18-20: Statement “Including 33, 36, 42, 29 … to 35, 37, 43 and 29 healthy controls” not clear. Consider rephrasing.

We have corrected the writing problem and modified the methods section in the abstract as follows: A total of 1563 pregnant women in Peking University Third Hospital from February 2021 to October 2021 were enrolled. Serum samples were collected from patients 1 to 3 times during different trimesters. Among 102 singleton pre-eclampsia patients, we have collected samples from 33 patients in the first trimester (6-13 GW), 33 in the second trimester (14-28 GW), 41 in the third trimester (29-41 GW) and 28 after onset of pre-eclampsia. Samples from the first, second and third trimesters were collected before the onset of pre-eclampsia. Then we have collected 35, 37, 43, and 25 samples from 124 healthy pregnant women by matching age, BMI, and gestational weeks to the preeclampsia group, respectively. Serum levels of MMP-2, -7, -9 were detected by ELISA. The receiver operating characteristic (ROC) curve was used to evaluate their predictive value.

Introduction

Line 39: Change “Matrix” to “matrix”.

Line 45: Insert a reference for Gross’s work

Line 60: Change “plays” to play.

We have revised the writing problem and added a reference for Gross’s work

Results

Table 3: Authors should indicate the p-values obtained after the ROC curve analysis.

Lines 129-131: Remove underlines from footnotes under Table 1.

We have indicated the p-values obtained after the ROC curve analysis. (Table 3) In the meantime, I have removed underlines from footnotes under Table 1.

Major Comments

Results

Lines 169-170: Authors indicated that after the onset of PE, the MMP-9 level in PE group [1066.92 169 (829.73,1305.75)] ng/ml was significantly lower than that in control [806.00 170 (581.47,1051.00)] ng/ml, (P<0.05). This contradicts the values obtained and findings from Fig 1.

The wrong word is used in this sentence and We have changed "lower" to "higher".

Discussion

Lines 258-261: Authors should clarify the MMP-9 level in PE group and controls as suggested above, and consider rephrasing statements in the discussion.

All the above questions have been revised in the manuscript.

Sincerely,

Ayibaota Bahabayi.

Reviewer 2 Report

General concept comments:

-"PE" can represent either "pre-eclampsia" or "pulmonary embolism". PET (pre-eclamptic toxemia) may be easier to use

-Background usually describes the context of the study

-Font size differs throughout the paper and it is irritating to note

-It is very difficult to work out from the abstract how many samples were analysed in this study

-It is also unclear if samples were taken from women in the first trimester who subsequently developed pre-eclampsia or if women developed pre-eclampsia in the first trimester (not plausible).

There lacks significant clarity on whether this study is in fact a longitudinal study as stated by the authors in the abstract and on lines 208 and 266. The authors state that 106 pregnancies resulted in a PET outcome and there are 140 cases (33, 36, 42, and 29) of PET total. Were some of these patients longitudinal? How many were captured from first trimester onwards? If not all women were captured from first trimester, I do not think the authors can claim this as a strength of their study.

There lacks any discussion on the current use of biomarkers for the prediction/diagnosis of PE (PlGF/sFlt-1). Why is it important to find new biomarkers for PE? How would MMP markers compare to the already available tests? How would a prediction biomarker change the way PE patients are treated? This is a major limitation to the manuscript as written at the moment.

Statistically, the authors could have extracted more from their dataset. A combination of their demographic data and MMP values/ratios and a logistic regression model could significantly improve AUC’s and predictive values.

 Even though the dataset is small, and already identified as a limitation by the authors (line 277), I would have liked more insight within the text/tables on the percentage of PET patients with severe PE. In addition, some discussion on term/preterm PET would have strengthened the manuscript. From table 1, it seems that the PET cases skewed towards term PET and I wonder if there would be any drastic changes in MMP values between term and preterm.

  While I believe there is merit and potential to this work, I believe the manuscript lacks clarity in the study design and the authors have not extrapolated in depth the potential of their results. A significant amount of demographic/maternal characteristic data has been collected and while some is reported in the manuscript, it is not used to determine/spot trends in the results. In addition, there is conflict between what is said and what is shown: for example, women with autoimmune diseases were excluded, yet there are women with SLE included in Table 1

Specific comments: Case numbers (33, 36, 42, and 29) need to be communicated more clearly, especially in the abstract as this causes unclarity and confusion. Please see comment above on clarifying if all cases were longitudinally followed. There are many instances where the author writes “studies have shown” and only provide one reference or none at all (studies requires more than one reference to back up your statement). For example:Lines 39-40 – only one reference and this is a book chapter. Lines 66-68 – no references to studies that only measure MMP in one trimester (same for line 266-268). Referencing:  Over relying on reference #6. Most of the introduction seems to play around this particular reference. Lack of referencing in the discussion – not a single reference between 228 and 264. The discussion should encompass how your results fit into the current literature. Back up statements of MMP placental detachment, MMP destroying vascular integrity etc. with references, as your study does not prove this specifically.  Line 44-45 – no reference for Gross. The manuscript would benefit from a CONSORT diagram to explain the study design.   Sample collection (line 100-103): How was the serum separated from whole blood (centrifugation speed)?   Line 106-107: Kit lot number is listed, not catalogue number. Catalogue number is needed for purchasing and tracking which kit was ordered. More clarity in sentences used for statistical analysis. Mann Whitney for MMP values in PE vs Control, Krustal Wallis for MMP values in different trimesters (plus after onset). However, if this is truly a longitudinal study with repeated, dependent, variables in each trimester group then a Friedman test for nonparametric data is more appropriate. Statistical analysis only mentions analysis of MMP. How were maternal characteristics analysed?   Table 1: Include ethnicity if this was recorded; what do you mean by delivery time? this is completely unclear: did you mean gestational age? Can SD be used instead of M[IQR] for blood pressure measurements? From table 1 there doesn’t seem to be a major significant difference in the BP measurements of your PE group in SBP, which you would expect. ACOG (as you have listed as your definition criteria) definition of PE is SBP over or equal to 140 and DBP over or equal to 90. This does not seem to be reflected in the clinical characteristics.  Please use actual p values in figure 1 rather than stars of significance.    Section 3.3: PPV and NPV should be included within text as PPV is an indication of the rule in capacity of MMP values and NPV rule out. Please include 95% CI for all values, not just AUC.

Round 2

Reviewer 2 Report

There are several points raised by the reviewers that are answered but these answers are not included in the text. Suggest updating the text to reflect all points raised

Author Response

Dear Reviewer,

The title of the manuscript is Expression of matrix metalloproteinase-2、-7、-9 in serum during pregnancy in patients with pre-eclampsia: a prospective study. Thank you very much for your modification suggestions. The following is the modification I have made in response to your modification comments.

New manuscript further describes our experimental design in detail, and further complements the results and discussion section.

For data, we added corresponding blood-pressure data for each trimester and the number of patients in each subgroup (Table 1). At the same time, a flow chart is drawn to better explain our experimental design.

Sincerely,

Ayibaota Bahabayi.